# Caregiver Burden Is Reduced by Social Support Services for Non-Dependent Elderly Persons: Pre-Post Study of 569 Caregivers

**DOI:** 10.3390/ijerph192013610

**Published:** 2022-10-20

**Authors:** Sylvie Arlotto, Stéphanie Gentile, Alice Blin, Anne-Claire Durand, Sylvie Bonin-Guillaume

**Affiliations:** 1Service d’Evaluation Médicale, Assistance Publique Hôpitaux de Marseille, 13005 Marseille, France; 2Health Service Research and Quality of Life Center (EA 3279), School of Medicine, La Timone Medical Campus, Aix Marseille University, 13005 Marseille, France; 3Neurosciences of Systems UMR-Inserm 1106, Aix Marseille University, 13005 Marseille, France; 4Internal Medicine and Geriatric Department, Hôpitaux Universitaires de Marseille, Assistance Publique Hôpitaux de Marseille, 13009 Marseille, France

**Keywords:** family caregiver, burden, non-dependent elderly person, social support services

## Abstract

Background: Informal care provided by family caregivers (FCGs) to elderly persons is associated with a high risk of burden and poor health status. Social support services (3S) for the elderly persons were characterized by assistance in various activities of daily living. This study aimed to analyze the impact of 3S on the burden of FCGs of elderly persons living in the community and identify factors associated with changes in their burden. Methods: This pre-post study was performed in the southeast of France: FCGs of non-dependent elderly persons still living at home who received a 3S were consecutively included. FCG burden was assessed with the Mini-Zarit scale before the setting up of the 3S (pre-3S) and 6 months after (post-3S). Results: A total of 569 FCGs were included in the study. Mean age of the FCGs was 62.9 years old (±13.3), 67% were women, 61.2% were children or stepchildren. Burden was present for 81% of FCGs. In most cases, 3S targeted household chores (95.8%); 59.8% of elderly persons and their FCGs were fully satisfied. The improvement in burden was greater for FCGs perceiving less obstacles post-3S in helping elderly persons (OR = 4.083) but also for FCGs fully satisfied with the 3S (OR = 2.809) and for FCGs whose perceived health status had improved post-3S (OR = 2.090). Conclusions: FCGs of non-dependent elderly persons experience a burden similar to those of dependent elderly persons. The implementation of a 3S in daily life helps to reduce their burden.

## 1. Introduction

If recent demographic trends persist, in 2060, almost 30% of the French population will be aged over 60 [1]. In this context, promoting quality of life, well-being and dignity is a challenge [2]. Recent data have shown that about 20% of the European population over 65 years of age is to some extent dependent and, for the most part, assisted by a family caregiver (FCGs), mainly women [3]. These FCGs are essential to relay care and help to elderly persons and will be increasingly involved in the coming years [4]. The economic value of this unpaid informal care has been estimated at between 6.1 and 8.3 billion euros per year in France [5].

In France, between 8 and 11 million persons regularly help one or more persons for health or disability reasons (i.e., nearly one out of six persons) [6]. In 2015, 3 million persons aged 60 and over and living at home regularly received assistance for a health problem or disability [7]. The FCG’s profile varies according to the age of the person being cared for. With advancing age, the help provided by spouses (80% when assisted persons lived in a couple) decreases in favor of help provided by children [6].

The type of care provided by FCGs varies greatly, from nursing and care management to everyday activity assistance. When the help provided exceeds the physical and mental capacities of the FCG, it becomes a chronic stressor generating FCG’s burden [3].

For health workers, it is important to correctly identify the most involved FCGs, assess the level of stress associated with caregiving, and create a partnership with the FCGs [8]. Chronic stress has a negative effect on the FCG’s mental and perceived health and increases the risk of chronic health problems and the development of new diseases [9]. In addition, the stress and emotional distress of FCGs has serious consequences on their progressive inability to fulfill their role as FCGs, resulting in early institutionalization or unplanned hospitalization of the elderly person [10] or the risk of mistreatment for the care recipient [11,12]. To prevent/avoid these situations interventions and provided support are necessary [3,9,13].

Two main factors contribute to reducing the burden of FCGs. The first is to receive additional help from another family member [14]. The second is to benefit assistance from institutional system [6]. In France, the Caisse d’Assurance Retraite et de Santé Au Travail (CARSAT: national administration for retirement and occupational health) allocates assistance for non-dependent old beneficiaries who apply for social support. The CARSAT is responsible for pension management and health and safety at work, and provides assistance to vulnerable persons. This national institution is represented in all French regions. For the most dependent elderly persons, aid is provided by another regional institution. The South–East CARSAT (for the Provence Alpes Côte d’Azur region, with more than 1 million people aged 65 and over, out of 5 million inhabitants) has implemented Social Support Services (3S), a formal service developed to provide social support, e.g., household chores, ironing, shopping, or travel support, to retired beneficiaries who are not yet dependent but who are at risk of becoming so. After assessment by a social worker, support is proposed to retired beneficiaries who are self-sufficient but need logistical assistance in their daily lives [15]. However, CARSAT does not fully cover the costs associated with the support: the level of CARSAT coverage is correlated with the income level of the beneficiary. The higher the income, the lower the share of CARSAT coverage. Thus, the financial contributions of elderly persons vary from 10% to 73% of the 3S cost. We hypothesized that a specific and personalized support for old beneficiaries would improve the burden of their FCG.

The main objective was to analyze the impact of 3S on FCGs burden and identify factors associated with changes in FCGs burden.

## 2. Materials and Methods

### 2.1. Study Design

A pre-post study was conducted between 2016 and 2017. This study compared the caregiver burden level before the implementation of the 3S (pre-3S) and the burden level 6 months after its implementation (post-3S).

### 2.2. Study Population

The population consisted of FCGs caring for people aged 70 and over who were living at home, not dependent according to the AGGIR grid (Autonomie Gérontologie Groupe Iso Ressources), which categorizes persons according to their dependence for activities of daily living [16], without serious chronic illness and CARSAT beneficiaries.

To be included, the FCG had to have been designated by the elderly person who had applied for 3S at the South-East CARSAT.

The FCGs were consecutively included between 1 April 2016 and 30 June 2017 and were followed for 6 months (post-3S). If the follow-up after 6 months was not possible, FCGs were excluded from the study.

### 2.3. Social Support Services Setting Up

When the elderly person applied for the need of assistance, a CARSAT social worker went to the elderly person’s home to assess his or her health, frailty and needs according to the usual procedure. Following this assessment, a 3S was offered to the elderly person. Once the latter has accepted the 3S, CARSAT must select the provider who will carry out the 3S findings. There is a dependent remainder for the elderly person depending on his/her income.

### 2.4. Data Collection

In addition, social workers, trained in the study procedures, distributed a self-assessment questionnaire to the FCG. The FCG had previously been invited, through an information letter, to be present at the time of the elderly person’s assessment.

If the FCG could not be present, he/she was then contacted by telephone by a trained clinical researcher to complete the questionnaire.

The 6-month follow-up was conducted solely by a telephone call from the clinical researcher. The FCG were considered “unreachable” after 10 unanswered phone calls.

### 2.5. Measurement

The burden of FCGs has been evaluated by the Mini-Zarit scale [10,17]. The Mini-Zarit scale consists of 7 items seed bellow, scored in 3 points Likert scale from 0 (never), 0.5 (sometimes) to 1 (nearly always). The 7-items abbreviated version of the original Zarit Burden Interview (ZBI) [17] is one of the abbreviated versions with the best metrological qualities with a Cronbach’s alpha between 0.86 and 0.96 depending on the FCG profiles [18].

Does helping for the care recipient result in difficulties in your family life?Does helping for the care recipient result in difficulties in your relationships with friends, in your hobbies, or in your work?Does helping for the care recipient result in an impact on your health (physical and/or psychological)?Do you feel that you no longer recognize your care recipient?Are you afraid for your care recipient’s future?Would you like (more) help to take care of your care recipient?Do you feel burdened taking care of your care recipient?

The total score ranges from 0 to 7. Scores greater than 1 indicate a burden. The higher the score, the greater the burden.

### 2.6. Other Data Collected

Other variables collected are presented in Table 1.

### 2.7. Data Analysis

Statistical analysis was conducted using SPSS Statistics version 20 software (IBM Corp., Armonk, NY, USA ). All variables were examined through descriptive analysis. Qualitative variables were described by their frequencies and percentages and quantitative variables by their mean and standard deviation (±), minimum, median, and maximum.

A pre-post analysis was also conducted through matched-statistics to evaluate the impact of the 3S 6 months later: Cochran’s Q Test for binary variables (perceived health, difficulties and frailty) and the Wilcoxon or Student Test for quantitative variables (Mini-Zarit score). The associations between qualitative variables were measured by the Chi^2^ test and the exact Fischer test for small numbers.

For each FCG we created several qualitative variables to categorize the evolution of burden, perceived health, and perceiving obstacles. The response modalities for these variables were binary: “improvement or not” of the measured variable.

A multivariate logistic regression model was then applied to identify factors significantly associated with burden improvement. The binary variable “burden improvement or not burden Improvement” was tested in univariate analysis with all the other variables collected (Table 1) and those for which the *p*-value was less than 0.2 were included in the logistic regression after checking the correlations.

## 3. Results

Of the 876 FCGs interviewed at the time of inclusion, follow-up was possible for 569 (65%) of them. For the 307 that were not included in the study, reasons were: refusal of the 3S by the elderly person (38%), refusal to answer the questionnaire (21.5%), unreachable (34.5% of the cohort), and death of the elderly person (6%). The demographic characteristics of the FCGs not included were similar to those of the FCGs included. They were 62 years old on average (*p* = 0.557), 60% were retired (*p* = 850), and one-third were spouses (*p* = 0.620).

### 3.1. Characteristics of the FCGs Included in the Study (Pre-3S)

The characteristics the 569 FCGs are presented Table 2. The average age was 63 years (± 13.3), 67% of the FCGs were women and 79.2% of the FCGs lived as a couple. Most of the FCGs were children or stepchildren (61.2%); spouses represented 27.6%. The others (11.2%) were mostly (71.7%) family members (grandchildren, and siblings), the rest were friends or neighbors.

The child FCGs were younger (56.3 years ± 7.7 vs. 79.2 ± 6.1 for spouse FCGs and 59.5 ± 15.8, *p* < 0.001 for other FCGs) and were more often women (75% vs. 43.3% of spouse FCGs, *p* < 0.001).

Moreover, child FCGs had a higher level of education (61.2% vs. 21.1% of spouse FCGs, *p* < 0.001) and were more often employed (58.9% vs. 0.6% of spouse FCGs, *p* < 0.001).

Among all FCGs, 13.7% reported having another care recipient, children excluded. This was more common among child FCGs (17.5% vs. 4.5% of spouse FCGs, *p* < 0.001).

More than half of FCGs reported “feeling supported by another family member”, especially child FCGs (61.2% vs. 40.1% of spouse FCGs, *p* < 0.001).

The average Mini-Zarit score was 2.75 (± 1.6), with child FCGs feeling the highest burden (2.9 ± 1.6 vs. 2.6 ±1.6 for spouse FCGs and 2.2 ± 1.4 for the other FCGs, *p* < 0.001). In addition, burden was present for 81% of FCGs (i.e., Mini-Zarit score > 1) pre-3S setting up. The 19% of non-burdened FCGs were predominantly “others FCGs”, such as neighbors or grandchildren (19.4% among FCGs without burden vs. 9.3% among FCGs with burden, *p* < 0.05).

More than half of the FCGs perceived obstacles in helping the elderly person, especially spousal FCGs (63.7% vs. 49.1% of child FCGs, *p* < 0.05).

In addition, 89% of the FCGs were bothered to provide support to their old care recipient. The most common reason was their own health status (82.8% of spouse FCGs vs. 29.6% of child FCGs, *p* < 0.001), the lack of time (59.8% of child FCGs vs. 8.3% of spouse FCGs, *p* < 0.001) and their family obligations, for example, lack of time available for children (40% of child FCGs vs. 2.5% of spouse FCGs, *p* < 0.001). Other reasons for their worries were the lack of humans or financials resources, the lack of expertise of professionals, the lack of dialogue with professionals or support services and the lack of information.

Nearly half of the FCGs had difficulty for having a few days break because of the need of help provided to the elderly person (54.9% of child FCGs vs. 37.6% of spouse FCGs, *p* < 0.001), but also in their daily life (52.6% of child FCGs vs. 38.9% of spouse FCGs, *p* < 0.005). For a quarter of the FCGs, assistance had a negative impact on their relationships with other family members (33.9% of child FCGs vs. 13.4% of spouse FCGs, *p* < 0.001).

The majority of FCGs reported a good perceived health, especially child FCGs (80.2% versus 44.6% of spouse FCGs, *p* < 0.001). In contrast, one-quarter of the FCGs felt anxious and one-third complained of sleep disorders. In addition, one in ten FCGs felt lonely.

Finally, in terms of frailty, only one-third of FCGs were rated as robust by the FiND questionnaire assessment (44% of child FCGs versus 14.6% of spouse FCGs, *p* < 0.001).

### 3.2. Characteristics of the 3S and Level of Satisfaction

In most cases, 3S targeted household chores (95.8%). Then, for nearly one elderly person in five, the 3S consisted of assistance for shopping/groceries (17.2%) or ironing (16%). The other types of assistance included assistance during outings (6.3%), help with meal preparation (6.2%), help for grooming (2.3%), help with meals (1.9%), help for administrative tasks (1.9%) and finally help for dressing (0.9%).

One in five elderly persons received two forms of assistance and 10% received more than two. The most common type of assistance was combining household chores with ironing (29%) or combining household chores with shopping/groceries (27%).

Nearly two-thirds of elderly persons and their FCGs felt that their needs were fully met (59.8%) by the 3S.

The causes of dissatisfaction included the desire to receive more hours of assistance (70.7%) and 19 FCGs (8.3%) felt their care recipient should have received other types of support.

### 3.3. Impact of the 3S (Post-3S)

Post-3S, burden was present for 56.2% of FCGs. Furthermore, 61.7% of the FCGs had a lower score, 20.2% had an equal score, and 18.1% had a higher burden score. The average Mini-Zarit score decreased (2.75 ± 1.6 pre-3S vs. 1.9 ± 1.6 post-3S, *p* < 0.001). Analysis of the items making up the Mini-Zarit Scale (Table 3) shows improvement in all components of the burden (*p* < 0.05).

At post-3S, 52% of FCGs reported less need for help and the percentage of FCGs who experienced difficulties in their family life helping the elderly person decreased (53.3% pre-3S vs. 23.6% post-3S, *p* < 0.001).

Thus, 3S had no impact on perceived health status (70.1% of FCGs reported a good perceived health pre-3S vs. 72.1% post-3S) and frailty of the FCGs (37.1% of robust FCGs pre-3S vs. 37.3% post-3S).

Of the 108 FCGs without burden pre-3S, only 26 (24%) had developed burden post-3S. This burden was mostly related to worrying about their care recipient’s future (n = 16).

### 3.4. Analysis of Factors Contributing to Burden Improvement

The results of the univariate analysis of factors related to burden improvement post-3S are presented in Table 4.

According to the multivariate analysis (R2 = 0.244), FCGs perceiving fewer obstacles in helping elderly persons post-3S were four times more likely to improve their burden score (OR = 4.083, 95% CI 1.318–4.525). Similarly, FGCs who were fully satisfied with the 3S were three times more likely to have reduced their burden score (OR = 2.809, 95% CI 1.899–4.154). FCGs with another care recipient two fold more improved their burden (OR = 2.442, 95% CI 1.318–4.525). Finally, an improvement in burden was observed for FCGs whose perceived health status had improved post-3S (OR = 2.090, 95% CI 1.211–3.606) and for those who felt anxious pre-3S (OR = 1.715, 95% CI 1.093–2.691).

## 4. Discussion

A major strength of this study was the finding that 81% of the FCGs of non-dependent elderly persons have a burden. In addition, our results showed the decrease in this burden for 73% of FCGs after the setting up of the 3S. However, 3S did not show a direct effect on the perceived health or the frailty of the FCGs.

This study is original in that of it is targeting FCGs of elderly persons without serious chronic diseases, autonomous, and living at home. To our knowledge, no study has targeted these FCGs, who are the most numerous and constitute a real public health challenge, nowadays and in the future [20,21]. Moreover, while most studies have focused on psychological or medical support activities addressed to FCGs [22], our study exclusively concerns the impact on FCGs of 3S addressed to their care recipient. At last the large sample and the little proportion of persons lost post-3S constitute a good level of internal validity.

The content of the 3S was tailored by the social worker according to the needs of the elderly persons after the in-home assessment. The 3S was mainly characterized by the implementations of assistance for household chores, it could potentially cover a wide range of areas, such as assistance with activities of daily living, home adaptation work, a tele-alarm system, prevention workshops or a respite stay for the elderly person. Yet, France has developed a wide range of social services to help disabled and not disabled elderly persons, according to their needs which has proven to be an effective strategy to reduce unmet needs of those beneficiaries [23]. Before being implemented, the 3S had to be accepted by the elderly person. We only assess the impact of the 3S validated by the elderly person. The choices were primarily focused on the management of daily living. The remaining costs to the elderly person may justify not selecting the other proposed types of support. Even though, most of the dyads were satisfied; the reasons for dissatisfaction were remaining unmet needs, maybe underlying financial difficulties to accept more.

The 3S appears to positively impact the perception of obstacles in helping the elderly person (improved for more than a half of the FCGs). However, not for other areas. Concerning employed FCGs, even the assistance provided to a family member affects job stability [24] and there is a link between the intensity of assistance and employment [25], the results do not highlight any significant improvement in the burden on employed FCGs following the implementation of the 3S. Indeed, the intervention of professionals does not exactly mean a decrease in the assistance provided by the FCG but rather a change in the form of assistance provided to the care recipient [26].

With the 3S, the perceived health and the frailty levels improved only for one in five. These results are encouraging as the main purpose of the 3S is to bring social assistance to the care recipients, and aiming at indirectly input some improvement on FCGs health status [27].

According to a literature review, the burden includes several components [28]: a psychological component related to the difficulties experienced and an emotional component related to the relationship with the elderly person, and finally a physical component related to the type and number of tasks performed. Additionally, although burden is a central concept in this study, it was not explored with the Zarit Burden Interview [17] but with the 7-item Mini-Zarit scale. According to Higginson et al. the reduced 7-item version offers the best metrological qualities when a shortened version is needed, while exploring the different dimensions of burden [18].

The implementation of the 3S has impacted the “Physical” component of the burden. However, the whole process of implementing the 3S may also have a positive impact on the psychological and emotional components of burden. Indeed, two Mini-Zarit items: “fear of the future” or “feeling of no longer recognizing their care recipients”, showed a significant improvement after the 3S implementation. Additionally, the multivariate analysis revealed a greater decrease in the burden for FCGs that felt anxious before the PPS implementation. Additionally, at last, FCGs who perceived an improvement in their health status experienced also a greater improvement in their burden referring to the emotional component of the burden [29].

Surprisingly [3], the results show that about a quarter of FCGs who had no burden before the setting up of the 3S developed burden after the 3S. We believe that this may be related to the 6-months delay following the setting up of the 3S; time during which the elderly person’s health status may have deteriorated.

Our study has several limitations. First, we only included elderly persons who asked for social support in daily living tasks who were independent but mostly frail [15]. A second limitation regarding the measurement is the subjectivity bias of declarative health data. As a result, comparisons of pre- and post-3S health status were based solely on perceived health. Additionally, the single measure of loneliness, anxiety or sleep disorders did not allow a comparison of these variables before and after.

In addition, some data were collected by filling out a self-questionnaire by the FCGs, and others were collected during telephone interviews. The use of these two methods therefore represents a measurement bias.

Finally, the delay between the two questionnaires regarding the effective implementation of the 3S might have been, indeed, less than six months. Thus, the measurement of the effects of the 3S could be carried out slightly prematurely in certain cases with the lack of significance for some results.

Thus, these results cannot be generalized to every non-dependent elderly persons living at home. The recent COVID-19 pandemic have strengthened the role of FCG due to the lockdown and the barrier measures toward elderly persons at high risk of a severe form of the disease [30]. In some cases, home help was cut down to avoid the risk of contamination. Yet, the FCG were on the bridge for the caring of their care recipients. That led to an exacerbation of the levels of stress and burden. This unpredictable and worldwide situation has emphasized the role of FCG in our health care system with a risk on their own psychosocial and health negative issues. Thus, the lack of consideration of FCGs has important implications for the health care system [14,31,32].

This context then underlines the need to take into consideration the role of the FCGs [33,34] and the obstacles they are facing [32], with regard to their physical and mental health. The 3S provides answers for some social issues encountered by FCGs, but not for all of their concerns.

From a global health and social perspective, FCG are at risk of adverse care-related outcomes and must benefit from support in maintaining their own health as well as their caregiving responsibilities [35].

The first action should be to identify FCGs during routine medical consultations. Then, propose a dedicated consultation as already done for some chronic disease [36] before physical and/or psychological disorders appear [37,38]. By a global approach, medical, psychological, emotional and social needs evaluation, this consultation would advise and refer the FCG to the most appropriate organizations and associations [39].

## 5. Conclusions

Our results show that the population of FCGs of non-dependent elderly persons is very similar to that of FCGs of dependent elderly persons or persons suffering from serious diseases in terms of socio-demographic characteristics, but also in terms of burden level, perceived health status, and risk of frailty. The evaluation of the implementation of a social assistance plan to improve home support for elderly persons shows that the implementation of 3S improves the burden on FCGs.

Health professionals and/or social workers must appropriately identify FCGs to offer care adapted to their needs and expectations and not only through the needs of the person being cared for. The implementation of dedicated consultations could help FCGs discover the limits of their role and to evaluate the difficulties they encounter so that they can be supported.

## Figures and Tables

**Table 1 ijerph-19-13610-t001:** Variables collected pre and post-3S.

Variables Collected from FCGs	Pre-3S	Post-3S
Sociodemographic data		
Age, gender, family status (living alone or as a couple) and number of children living at home	√	
The relationship with the elderly person Spouse FCGs (spouses, partners, and ex-spouses) Child FCGs (children and stepchildren) Other FCGs (no family relationship and other family members)	√	
The level of education, grouped into 2 categories: above or below High school education	√	
Professional activity grouped into two categories: Employed FCG/Non-Unemployed FCG	√	
FCG with another care recipient (elderly subject, child,…) (yes/no)	√	
FCG with the support of another family member (yes/no)	√	
Difficulties in providing assistance		
Perceiving obstacles in helping the elderly person evaluated on a 5-point Likert scale ranging from “not at all” to “a lot” (recoded as: feeling obstacles YES (moderately, a lot, enormously)/feeling obstacles NO (a little, not at all).	√	√
FCG bothered to providing support by his/her lack of expertise, lack of time for caring, lack of humans or financials resources, family obligations, own health status, lack of dialogue with professionals or support services, lack of information.	√	
The consequences of the assistance provided		
Providing support, feeling a negative impact on relationships with other family members, on the daily life, on having a few days break.	√	
Perceived health (5 points on the Likert scale from “excellent” to “bad”), (Recoded as: good health YES (good, very good, excellent)/NO (poor, bad)	√	√
FCG complaining of sleep disorders (5 points on the Likert scale)	√	
FCG feeling lonely (5 points on the Likert scale)	√	
FCG feeling anxious (5 points on the Likert scale)	√	
Frailty, evaluated by the “Frail Non-Disabled” (FiND) self-questionnaire which consists of five questions: two are specifically aimed at identifying individuals with mobility disability, and three assess components of the frailty syndrome: weight loss, exhaustion, and sedentary behavior. It categorizes robust subjects from frail ones [19]	√	√
Characteristics of the 3S proposed by the social worker after home visit		
Checklist with several possible answers: assistance for household chores, ironing, shopping/groceries, outings, administrative tasks, meal preparation, taking meals, grooming, dressing, securing the house (shower, stairs…)		√
FCG satisfied or dissatisfied with the 3S proposed		√
Reasons for dissatisfaction		√

**Table 2 ijerph-19-13610-t002:** Profile of FCGs pre-3S.

	N = 569
Average age	63 ± 13.3
Women % (n)	67 (381)
Spouse % (n)	27.6 (157)
Child % (n)	61.2 (348)
Other % (n)	11.2 (64)
Living as a couple % (n)	79.2 (446)
High school education % (n)	49.4 (276)
Employed % (n)	40.1 (228)
FCG with another care recipient % (n)	13.7 (77)
FCG with the support of another family member % (n)	54 (307)
FCG bothered by the lack of information % (n)	3.5 (20)
FCG bothered by the lack of dialogue with professionals or support services % (n)	5.4 (31)
FCG bothered by their own health status % (n)	44.5 (253)
FCG bothered by their family obligations % (n)	30.8 (175)
FCG bothered by the lack of humans or financials resources % (n)	17.6 (100)
FCG bothered by the lack of time for caring % (n)	43.6 (248)
FCG bothered by the lack of expertise % (n)	13.7 (78)
FCG feeling a negative impact on the relationships with other family members % (n)	27.1 (154)
FCG feeling a negative impact on their daily life % (n)	48 (273)
FCG feeling a negative impact on having a few days break % (n)	49.6 (282)
FCG perceiving obstacles in helping the elderly person % (n)	53.3 (303)
Robust FCG % (n)	37.1 (211)
FCG feeling a good perceived health % (n)	70.1 (399)
FCG complaining of sleep disorders % (n)	32 (181)
FCG feeling lonely	9.3 (53)
FCG feeling anxious	26.3 (148)

**Table 3 ijerph-19-13610-t003:** Improvement of each items score of Mini-Zarit Scale post-3S.

Mini-Zarit Scale Items	FCGs for Whom the Item Improved % (n)
Does helping for the care recipient result in difficulties in your family life?	23.6 (134)
Does helping for the care recipient result in difficulties in your relationships with friends, in your hobbies, or in your work?	23.2 (132)
Does helping for the care recipient result in an impact on your health (physical and/or psychological)?	28.8 (164)
Do you feel that you no longer recognize care recipient?	17.2 (98)
Are you afraid for your care recipient’s future?	23.5 (134)
Would you like (more) help to take care of your care recipient?	52.2 (297)
Do you feel burdened taking care of your care recipient?	36 (205)

**Table 4 ijerph-19-13610-t004:** Evolution of the profile of FCGs according to the improvement of their burden.

	No Burden Improvement N = 218	Burden Improvement N = 351	*p*
Average age (years)	63.4 ± 12.7	62.7 ± 13.6	0.548
Women % (n)	64.7 (141)	68.4 (240)	0.362
Spouse % (n)	28.4 (62)	27.1 (95)	0.518
Child % (n)	61.5 (134)	61.0 (214)	
Other % (n)	10.1 (22)	12.0 (42)	
Living as a couple % (n)	77.9 (169)	80.1 (277)	0.535
High school education % (n)	49.8 (105)	49.1 (171)	0.886
Employed % (n)	38.5 (84)	41.0 (144)	0.555
FCG with another care recipient pre-3S % (n)	8.9 (19)	16.7 (58)	0.009
FCG with the support of another family member pre-3S % (n)	53.7 (117)	54.1 (190)	0.915
FCG bothered by the lack of information pre-3S % (n)	2.8 (6)	4.0 (14)	0.436
FCG bothered by the lack of dialogue with professionals or support services pre-3S % (n)	6.0(13)	5.1 (18)	0.670
FCG bothered by their own health status pre-3S % (n)	46.3 (101)	43.3 (152)	0.480
FCG bothered by their family obligations pre-3S % (n)	28.9 (63)	31.9 (112)	.449
FCG bothered by the lack of humans or financials resources pre-3S % (n)	15.1 (33)	19.1 (67)	0.229
FCG bothered by the lack of time for caring pre-3S % (n)	38.1 (83)	47.0 (165)	0.037
FCG bothered by the lack of expertise pre-3S % (n)	13.3 (29)	14.0 (49)	0.825
FCG feeling a negative impact on the relationships with other family members pre-3S % (n)	24.3 (53)	28.8 (101)	0.244
FCGs feeling a negative impact on their daily life pre-3S % (n)	43.1 (94)	51.0 (179)	0.067
FCG feeling a negative impact on having a few days break pre-3S % (n)	47.2 (103)	51.0 (179)	0.384
FCGs perceiving fewer obstacles in helping the elderly person post-3S % (n)	34.9 (76)	69.2 (243)	<0.001
FCGs more robust post-3S % (n)	12.8 (28)	18.8 (66)	0.063
FCGs with improved perceived health status post-3S % (n)	11.0 (24)	21.4 (75)	0.002
FCGs totally satisfied with 3S % (n)	45.4 (99)	68.7 (241)	<0.001
FCGs complaining of sleep disorders pre-3S % (n)	29 (63)	33.9 (118)	0.227
FCGs feeling lonely pre-3S % (n)	7.8 (17)	10.3 (36)	0.327
FCGs feeling anxious pre-3S % (n)	22.7 (49)	28.5 (99)	0.126

## Data Availability

The datasets used and/or analyzed during the current study are available from the corresponding author on reasonable request. These data will be used in other publications.

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
