# Peer review of "Caregiver Burden Is Reduced by Social Support Services for Non-Dependent Elderly Persons: Pre-Post Study of 569 Caregivers"

_ijerph, 2022, doi:10.3390/ijerph192013610_

Round 1
Reviewer 1 Report
I read with interest this study exploring the impact of Social Support Services on the burden of family caregivers of older adults. The objectives, introduction and methods are clear and the results are overall well presented. Discussion is based on the study results and limitations are recognised. However, there were numerous grammatical and wording mistakes and I highly recommend to have proofreading done to improve the language use in the whole manuscript. Some other minor points are made below.
Introduction
I would suggest to use ‘elderly person’ instead of ‘old person’.
Method
How were participants recruited? What did they have to do in order to participate in the study (e.g. provide informed consent)?
Please also mention what were the exclusion criteria of participants.
Did the CARSAT social worker collect data both from the elderly person seeking help as well as from the caregiver?
On page 3, in ‘Modality of data collection’ please change it to ‘Data collection’. Also, please change ‘he/her’ to ‘he/she’ in the same section.
On page 3, please change ‘Judgment criteria’ to ‘Scales’ or ‘Measurement’.
Please fix grammatical errors under the paragraph ‘Judgment criteria’.
As a clarification, did the authors consider FCG’s having burden if they scored greater than 1 on the mini-Zarit scale?
Results
Can you please clarify what this percentage (71.7%) means in the results below: “The others (11.2%) were mostly (71.7 %) family members (grandchildren, and siblings), the rest were friends or neighbors.”
Have the authors considered presenting Table 2 visually (e.g. a line-column chart or similar)? At present, it is difficult to grasp the information from the table.
On page 6, please clarify that robustness of FCG’s was in terms of frailty in the following sentence, as it is unclear otherwise: “Finally, only one-third of FCGs were rated as robust by the FiND questionnaire assessment (44% of child FCGs versus 14.6% of spouse FCGs, p < 0.001).”
Table 3 could be presented in the results in the text or visually, as the questions were already mentioned in the Method section and appears to be repetitive here.
Discussion
I found it surprising that the majority of the caregivers were children. Please can the authors discuss how this can affect the results, as very often the caregivers are spouses?
I also would like to see authors’ thoughts in the manuscript about using the Mini-Zarit scale rather than the Zarit Burden Interview. What implications does it have when using a Mini-Zarit scale? Perhaps the authors can state why they chose the Mini version, rather than the full scale for their study.
Please can the authors state clearly in the Discussion what is the importance of their work and what does this study add to the overall understanding of the topic?
Author Response
Dear Referee,
We thank you very much for your attention to our newly titled manuscript "Caregiver burden is reduced by Social Support Services for non-dependent elderly persons: pre-post study of 569 caregivers". Many thanks for the quality of your comments which allowed us to improve the quality of our manuscript for its publication in the International Journal of Environmental Research and Public Health.
We considered each of your comments, we made the necessary corrections and clarifications requested. You will find below the tables presenting our answers point by point.
Yours sincerely,
For all the co-authors,
Dr Sylvie ARLOTTO

Reviewer 2 Report
Overall, the manuscript is well done and interesting. Suggestions for improvement:
Consider using more equitable terminology such as "older adult". "senior adult", or other term rather than "old person".
Section 2.4 appears to have an incomplete sentence or a misplaced heading "Data collected"
Manuscript needs to be proof read for minor errors in spacing, punctuation, etc.
Author Response
Dear Referee,
We thank you very much for your attention to our newly titled manuscript "Caregiver burden is reduced by Social Support Services for non-dependent elderly persons: pre-post study of 569 caregivers". Many thanks for the quality of your comments which allowed us to improve the quality of our manuscript for its publication in the International Journal of Environmental Research and Public Health.
We considered each of your comments, we made the necessary corrections and clarifications requested. You will find below the tables presenting our answers point by point.
The manuscript also presents all the modifications made for this minor revision.
We remain at your disposal for any further information,
Yours sincerely,
For all the co-authors,
Dr Sylvie ARLOTTO

Reviewer 3 Report
Informal care given by family caregivers (FCGs) can place a burden on the FCGs. The study underlying the article examines to what extent this burden can be reduced by social support services (3S) like the implantation of assistance for household chores. The results of the pre-post study show, that a high proportion of the FCGs of non-dependent old persons have a burden and that this burden, especially the physical component of it, can be relieved by 3S. The improvement in burden was greater for FCGs feeling less obstacles in helping old person, for FCGs fully satisfied with the 3S and for FCGs whose perceived health status had improved post 3S.
The study is targeting FCGs of old persons without serious chronic diseases, autonomous and living at home. I am not aware of any studies on this particular group. The study is based on a clear research question, the design of a pre-post study seems appropriate to answer this question, the results, which are interesting and sometimes surprising, are critically discussed and the limitations are clearly stated.
I have two editorial questions:
Page 1 (last paragraph): „In France, between 8 and 11 million persons regularly help one or more persons for health or disability reasons (i.e. nearly one out of six persons in this age group)”. Which “age group” is meant here?
Page 3: The fourth paragraph ends with “Data collected”. What does that mean?
Apart from these two questions, I have no further queries.
Author Response

(The authors gave the same response as above.)
